# Maximizing LLMs Potential: Enhancing Mongolian Chinese Machine Translation with RL Agents and Adversarial Multi Knowledge Distillation

## Abstract

Despite the impressive performance of Large Language Models (LLMs) in Natural Language Processing (NLP), they still face challenges in low-resource translation tasks, particularly in Mongolian to Chinese machine translation, often yielding suboptimal results. To address this issue, we propose an innovative approach that combines multi-source knowledge distillation and incorporates Reinforcement Learning (RL) to help models acquire and transfer knowledge from LLMs more effectively. RL plays a crucial role in this, making dynamic decisions to determine useful information for low-resource translation models and how to extract it efficiently. We introduce a new reward function to comprehensively guide knowledge distillation, and experiments show that this approach harnesses the potential of LLMs, significantly improving translation quality in low-resource settings.

## 1 Introduction

As the field of natural language processing continues to advance, large-scale language models like GPT-3 (Brown et al., 2020) has shown excellent performance in a variety of NLP tasks. However, significant challenges remain in certain specific tasks, particularly in low-resource machine translation. A case in point is the translation from Mongolian to Chinese, where large-scale language models even underperform compared to Transformer (Dai et al., 2022; Zhu et al., 2023) (Table 1 presents the results of our comparison between the use of large language models and conventional models in Mongolian to Chinese machine translation) .

To address this issue, this study aims to introduce an innovative approach to improve low-resource machine translation from Mongolian to Chinese. Our method combines three key elements: first, extracting knowledge from large language models through multiple datasets and multi-source knowledge distillation; second, leveraging the dynamic decision-making capabilities of Reinforcement Learning (RL) agents to more effectively distill knowledge from large-scale language models. To achieve this, we propose a new reward function that comprehensively guides the knowledge distillation process. This function takes into account the intricate and diverse decision-making processes of both the student and teacher models during knowledge distillation. Finally, we introduce adversarial noise as a perturbation in the distillation process, and experimental results indicate that this significantly enhances the effectiveness of distillation.

In summary, the primary contributions of this research are as follows:

- By integrating multi-source knowledge distillation and reinforcement learning, we effectively optimize low-resource Mongolian-to-Chinese machine translation.
- We propose a new reward function specifically designed to guide the knowledge distillation process. It comprehensively considers multiple factors of the teacher model and the student model in distillation.
- The introduction of adversarial noise significantly improves the effectiveness of the knowledge distillation process.

Table 1: BLEU Scores in Mongolian-to-Chinese Machine Translation Across Different Language Models

| Method | BLEU |
|---|---|
| Transformer | 32.6 |
| Performer | 32.7 |
| Linformer | 31.3 |
| Llama2-13B (Touvron et al., 2023) | 29.1 |
| GPT-NEOX-20B (Black et al., 2022) | 28.2 |

## 2 RELATED WORK

### 2.1 KNOWLEDGE DISTILLATION IN NEURAL MACHINE TRANSLATION

Neural Machine Translation (NMT) has demonstrated superior performance across various translation benchmarks, establishing itself as the state-of-the-art method for translation. However, these models typically require large datasets for training, making them less practical in low-resource settings. Knowledge distillation emerges as a novel technology that not only compresses models but also delivers superior performance. Recent studies have shown that applying knowledge distillation to machine translation, especially in low-resource scenarios, can yield promising results. Firstly, Xu Tan et al. proposed a multilingual machine translation method using knowledge distillation (Tan et al., 2019). By using a single model as the teacher, they trained a multilingual model through knowledge distillation, effectively improving translation accuracy. Secondly, Wang focused on the selective application of knowledge distillation in NMT (Wang et al., 2021a). They introduced batch-level and global-level selection strategies to choose samples that are more suitable for knowledge distillation. Lastly, Jooste further explored the application of knowledge distillation in low-resource scenarios (Jooste et al., 2022). They used sequence-level knowledge distillation to train small student models on a simulated low-resource German-to-English translation task. These student models obtained knowledge from larger teacher models and ultimately outperformed them. Overall, these works comprehensively explore the combination of knowledge distillation and machine translation but do not consider the dynamics involved in the distillation process. We address this by introducing reinforcement learning to account for the dynamic changes between student and teacher models during knowledge distillation.

### 2.2 REINFORCED LEARNING KNOWLEDGE DISTILLATION

Chen's research (Chen et al., 2023) primarily addresses the issue of portfolio management in financial markets. By utilizing multi-agent reinforcement learning and knowledge distillation, it successfully creates effective investment strategies in complex financial environments. Wang's work (Wang et al., 2021b) introduces a novel method for multi-constraint molecular generation. By combining conditional transformers, knowledge distillation, and reinforcement learning, it effectively generates new compounds that meet multiple physicochemical constraints. Another study by Fan (Fan et al., 2021) proposes a reinforced knowledge distillation method based on policy gradient reinforcement learning, effectively tackling the issue of multi-class imbalanced classification. These works demonstrate the feasibility of agent-based knowledge distillation.

However, distinct from these previous studies, our current work introduces multiple teacher models and datasets. Furthermore, to adapt to the dynamic changes inherent in the distillation process, we introduce a brand-new reward function, as detailed in Section 3.2, making it applicable to NMT tasks.

## 3 METHODOLOGY

Our knowledge distillation architecture, illustrated in Figure 1, combines multi-source knowledge distillation with reinforcement learning to train a more powerful student model by amalgamating knowledge from multiple teacher models. During training, data generated by these teacher models

guides the student model while introducing adversarial noise to enhance its robustness. A reinforcement learning agent selects the optimal teacher model based on the current state's performance and adversarial loss, transferring its knowledge to the student model.

## 3.1 Adversarial Multi Knowledge Distillation

We utilize multi-source knowledge distillation to enhance the performance of the student model. Specifically, we train multiple teacher models with different architectures and objectives and distill their knowledge into a single student model.

Assuming we have $N$ teacher models and one student model, their output probability distributions are denoted as $M_t^i$ and $M_s$ respectively. The knowledge distillation loss $L_{\text{KD}}$ is composed of two parts: $L_{\text{similarity}}$ and $L_{\text{translation}}$.

$$L_{\text{KD}} = L_{\text{similarity}} + L_{\text{translation}} \tag{1}$$

$L_{\text{similarity}}$ is used to measure the similarity between the teacher models and the student model. We not only want the student model to learn the output probabilities of the teacher model, but, more importantly, we want the student model to capture a broader range of knowledge from the teacher model. Therefore, we combine cross-entropy loss and mean squared error (MSE) loss as a loss function to measure the overall similarity between the teacher model and the student model (Zhao et al., 2022).

$$L_{\text{similarity}} = \sum_{i=1}^{N} w_i \cdot \left( \text{MSE}\left(M_t^i, M_s\right) + \text{CrossEntropy}\left(M_t^i, M_s\right) \right) \tag{2}$$

$L_{\text{translation}}$ is used to measure the discrepancy between the student model's output and the ground truth data.

$$L_{\text{translation}} = \text{CrossEntropy}\left(M_y, M_s\right) \tag{3}$$

Additionally, we incorporate an adversarial loss $L_{\text{adv}}$ to further enhance the model's generalization capabilities:

$$L_{\text{adv}} = \max_{\delta} \left( \text{CrossEntropy}\left(M_s, M_t^i\right) - \lambda \cdot ||\delta||^2 \right) \tag{4}$$

where $\delta$ represents an adversarial perturbation, $\lambda$ is a hyperparameter that balances the trade-off between maximizing the adversarial loss and controlling the magnitude of the perturbation ($\delta$).

The overall loss function $L_{\text{total}}$ is a weighted combination of the above components:

$$L_{\text{total}} = \beta \cdot L_{\text{KD}} + (1 - \beta) \cdot L_{\text{adv}} \tag{5}$$

Here, $\beta$ is a hyperparameter used to balance the contribution of different loss components.

## 3.2 Novel RL Agent Knowledge Distillation

In order to make the student model pay more attention to the knowledge of the teacher model, we have made a crucial conceptual assumption by representing the iterative process of teacher models weighting as a Markov Decision Process (MDP) (Fan et al., 2021). $S$ represents the state space. $A$ is the space of teacher weights. The transition probability $T$ is deterministic, meaning that actions always transition $s$ to the next $s$ with a probability of 1. $r$ is the reward function. $r\left(s_t, a_t\right)$ denotes the reward received by the agent when taking action $a_t$ in state $s_t$.

Let $\mathscr{L}\left(\pi_\theta\right)$ be the objective function, which denotes the long-run average-reward under policy $\pi_\theta$. $S = \{s_t : t = 0, 1, 2, \ldots, T_{\max}\}$ denotes the state set of MDP. Mathematically, $\mathscr{L}\left(\pi_\theta\right)$ is defined as

$$\mathscr{L}\left(\boldsymbol{\pi}_{\theta}\right)=\sum_{\boldsymbol{s_t}\in\boldsymbol{S}}\boldsymbol{T}\left(\boldsymbol{s_t}\right)\sum_{\boldsymbol{a_t}\in\boldsymbol{A}}\boldsymbol{\pi}_{\theta}\left(\boldsymbol{a_t}\mid\boldsymbol{s_t}\right)r\left(\boldsymbol{s_t},\boldsymbol{a_t}\right). \tag{6}$$

To learn the optimal policy $\boldsymbol{\theta}^{\mathrm{opt}}=\underset{\theta}{\operatorname{argmax}}\mathscr{L}\left(\boldsymbol{\pi}_{\theta}\right)$, the gradient of $\mathscr{L}\left(\boldsymbol{\pi}_{\theta}\right)$ can be derived as:

$$\nabla_{\theta}\mathscr{L}\left(\boldsymbol{\pi}_{\theta}\right)=\nabla_{\theta}\sum_{\boldsymbol{s_t}\in\boldsymbol{S}}\boldsymbol{T}\left(\boldsymbol{s_t}\right)\sum_{\boldsymbol{a_t}\in\boldsymbol{A}}\boldsymbol{\pi}_{\theta}\left(\boldsymbol{a_t}\mid\boldsymbol{s_t}\right)r\left(\boldsymbol{s_t},\boldsymbol{a_t}\right) \tag{7}$$

$$=\sum_{\boldsymbol{s_t}\in\boldsymbol{S}}\boldsymbol{T}\left(\boldsymbol{s_t}\right)\sum_{\boldsymbol{a_t}\in\boldsymbol{A}}\nabla_{\theta}\boldsymbol{\pi}_{\theta}\left(\boldsymbol{a_t}\mid\boldsymbol{s_t}\right)r\left(\boldsymbol{s_t},\boldsymbol{a_t}\right) \tag{8}$$

$$=\sum_{\boldsymbol{s_t}\in\boldsymbol{S}}\boldsymbol{T}\left(\boldsymbol{s_t}\right)\sum_{\boldsymbol{a_t}\in\boldsymbol{A}}\boldsymbol{\pi}_{\theta}\left(\boldsymbol{a_t}\mid\boldsymbol{s_t}\right)\frac{\nabla_{\theta}\boldsymbol{\pi}_{\theta}\left(\boldsymbol{a_t}\mid\boldsymbol{s_t}\right)}{\boldsymbol{\pi}_{\theta}\left(\boldsymbol{a_t}\mid\boldsymbol{s_t}\right)}r\left(\boldsymbol{s_t},\boldsymbol{a_t}\right) \tag{9}$$

$$=E\left[\nabla_{\theta}\log\boldsymbol{\pi}_{\theta}\left(\boldsymbol{a_t}\mid\boldsymbol{s_t}\right)r\left(\boldsymbol{s_t},\boldsymbol{a_t}\right)\right] \tag{10}$$

where $r\left(\boldsymbol{s_t},\boldsymbol{a_t}\right)=E\left(\boldsymbol{r}_{t+1}\mid\boldsymbol{s_t},\boldsymbol{a_t}\right)$ denotes the single-stage expected rewards. $\boldsymbol{\pi}_{\theta}$ is a parameterized randomized stationary policy (composed of actions) with parameter $\boldsymbol{\theta}$, which maps each state $\boldsymbol{s_t}$ to a probability distribution $\boldsymbol{a_t}$. The update process of teachers weights can be formalized by Policy Gradient Reinforcement learning. The procedure of RLAMKD is presented as follows in 1

1. Initial Teacher Weights (Agent's Action):

$$\omega_i=\frac{1}{N} \tag{11}$$

$$\omega=[\omega_0,\omega_1,\ldots,\omega_N] \tag{12}$$

Where $N$ is the number of teacher models. As a result, each teacher model receives equal attention at the start, providing the agent with a fair starting point to evaluate the contribution of each teacher model.

2. Calculate the weighted loss of the teacher and student network: In knowledge distillation, we encourage the student model to learn from the teacher model by computing the following loss eq.(3).

3.Calculate the Reward $r_t$: The designed reward function is defined as follows:

$$r_t\left(\boldsymbol{s_t},\omega_t\right)=F_1(t) \tag{13}$$

$F_1(t)$ serves as a comprehensive measure of the model's performance over time. We aim for the agent to consider not only the current BLEU score of the model but also to acquire more knowledge from large language models. It is defined as follows:

$$F_1(t)=\int_0^T\left[-\mathcal{S}_{\mathrm{CRPS}}(t)-\mathrm{G}(t)-\mathrm{P}(t)+\mathrm{H}(t)+\alpha\cdot\mathrm{BLEU}(t)\right]e^{-\epsilon t}dt \tag{14}$$

Here, $\epsilon$ is a time-dependent discount factor used to adjust the present value of future rewards. $\mathcal{S}_{\mathrm{CRPS}}(t)$ represents the Continuous Ranked Probability Score, which measures the model's accuracy in probabilistic predictions and is minimized in this function. The Gini coefficient $G(t)$ quantifies the inequality or complexity among predictions made by teacher models. Consistency between the student and teacher models is captured by $P(t)$, which can be the Pearson correlation coefficient. $\alpha$ is a weight parameter used to control the importance of BLEU value in the reward function. Finally, $H(t)$, the Herfindahl Index, is used to gauge the diversity among predictions from teacher models. Appendix A.1 provides a detailed explanation and proof of the $\mathcal{S}_{\mathrm{CRPS}}(t)$ function.

4. Obtain State $\boldsymbol{s_t}$: The state $\boldsymbol{s_t}$ is composed of two features concatenated in the feature dimensions. The first feature is the output probability of the teacher network $\boldsymbol{p}_t^{\mathscr{T}}$. The second feature is the output probability of the student network $\boldsymbol{p}_t^{\mathscr{S}}$. The state $\boldsymbol{s_t}$ is represented as:

$$\boldsymbol{s_t}=\left[\boldsymbol{p}_t^{\mathscr{T}},\boldsymbol{p}_t^{\mathscr{S}}\right]$$

5. Update Actions of Policy Agent: The teachers weights $\omega_t$ correspond to the actions $\boldsymbol{a_t}$ of the policy agent. Policy gradient is employed in the policy agent network, and the loss function is defined as follows:

$$\mathscr{L}\left(\mathbf{s_t}, \boldsymbol{\omega}_t\right) = -\sum_{t=0}^{T_{\max}} r_t\left(\boldsymbol{s_t}, \boldsymbol{\omega_t}\right) \log \pi_\theta\left(\boldsymbol{\omega}_t \mid \mathbf{s_t}\right) \tag{15}$$

These two steps describe how to obtain the state $\boldsymbol{s_t}$ and how to update the actions of the policy agent, where the actions correspond to the teacher weights $\omega_t$. The policy agent network is optimized using policy gradient to maximize rewards and improve performance.

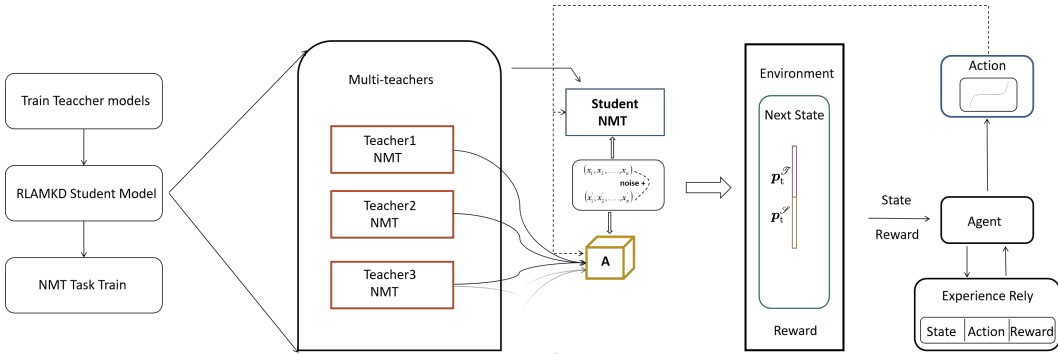

Figure 1: The complete architecture of the proposed RL Agent Multi-Source Knowledge Distillation method. The number of teacher networks can be modified based on the dataset.

## 4 EXPERIMENT

### 4.1 EXPERIMENTAL DATA AND CONFIGURATION

#### 4.1.1 CONFIGURATION

In this experiment, we set different learning rates for the teacher and student models, specifically 0.01 for the teacher model and 0.001 for the student model. Additionally, we utilized the Adam optimizer with these specified learning rates and implemented weight decay and gradual learning rate scheduling. The batch size was set at 64, and the maximum sequence length for the tokenizer was 128. All computations were conducted on a system equipped with NVIDIA A100 GPUs to ensure efficient training and evaluation. Each teacher model underwent training for 100 epochs on the respective dataset. Knowledge distillation training for the student model spanned 300 epochs. The entire training process extended over 200 epochs, with evaluations performed at the end of each epoch to calculate the BLEU score for the student model. To ensure a fair method comparison, we conducted hyperparameter tuning and selected the best settings based on the validation dataset results, with specific parameters $\beta = 0.7$, $\lambda = 0.1$ , $\epsilon = 0.01$ and $\alpha = 0.32$. For more training details, please refer to A.2.

We required three large-scale corpus datasets, namely English-Chinese, English-Turkish, and Russian-Chinese, to train the teacher models in order to distill the student model. For more details on the dataset information and the reasons for selection, please refer to A.3. Subsequently, our student model was trained and tested on Mongolian-Chinese, Korean-Chinese, and Japanese-Chinese translation tasks. Further information on these datasets can be found in A.4.

### 4.2 MODEL SELECTION

In our approach, we opt for the Transformer, Performer (Choromanski et al., 2020), and Linformer (Wang et al., 2020) as student models. These models have already demonstrated excellent performance in NLP tasks, particularly in machine translation, making them natural choices for our experiment. Specifically, we use the Transformer as the baseline model for this study to provide

a stable and empirically validated starting point. Additionally, this setup enables us to compare the effectiveness of different attention mechanisms during the knowledge distillation process and indirectly assess the potential of each mechanism.

Furthermore, we employ Llama2 and GPT-NEOX as teacher models for training. These are large-scale models that have been trained on extensive datasets, endowing them with strong text understanding and generation capabilities. Such large-scale training provides a rich and high-quality knowledge base for the student models. Going even further, these teacher models come from diverse research backgrounds and task objectives (Wei et al., 2022), each with its unique data distribution and specialization. This diversity offers a more comprehensive and multi-faceted knowledge base for the student models.

**Student Model Choose**

- Transformer: A standard Transformer(baseline).

- Performer: A strategy to improve the attention mechanism of Transformer through nonlinear mapping (Choromanski et al., 2020).

- Linformer: An improved strategy for Transformer's Attention Mechanism through Low Rank Matrix Approximation (Wang et al., 2020).

**Teacher Model Choose**

- Llama2: Llama2 is a large-scale pre-trained model released by Meta AI. It is part of a series of pre-trained and fine-tuned large language models (LLMs). According to the referenced work (Touvron et al., 2023), Llama2's model performance is evaluated with various parameter configurations, and the choice of a 13-billion-parameter model was made considering both model performance and hardware resource limitations.

- GPT-NEOX: GPT-NEOX is a self-regressive Transformer decoder model proposed by EleutherAI. Its architecture largely follows that of GPT-3 but incorporates some improvements. It is an open-source model that can to some extent replace GPT-3. According to the description in reference (Black et al., 2022), we have chosen its 20-billion-parameter model.

## 4.3 RESULT

Figure 2 (a) illustrates the relationship between the number of teacher models (Teacher Numbers) and their corresponding Bleu scores. As the number of teacher models increases, Bleu scores show an upward trend, indicating that increasing the number of teacher models can enhance performance. However, it's worth noting that after reaching a certain threshold of five teacher models, the improvement in performance becomes less significant, suggesting a saturation point. Furthermore, further increasing the number of teacher models does not lead to further performance gains, indicating stability within a certain range.

Figure 2 (b) compares two different models, Llama2-13B and GPTNEXO-20B, in the context of multilingual knowledge distillation with varying numbers of teacher models (Teacher Numbers). The results demonstrate that both models show increasing Bleu scores as the number of teacher models increases, consistent with the findings in Figure 2 (a). However, once a certain threshold of teacher model numbers is reached, Bleu scores stabilize.

Moving to Figure 3 (c), we analyze the effectiveness of training teacher models with different corpora for multi-source distillation. The results reveal that the "Mix" training method consistently achieves the highest Bleu scores in most cases, further corroborating that our multilingual approach contributes to improved translation performance to some extent.

Figure 3 (d) examines the relationship between Bleu scores and sentence length for OurKD and StKD (Standard KD). The results indicate that OurKD shows an upward trend in Bleu scores as sentence length increases, whereas StKD exhibits a declining trend in Bleu scores for longer sentences. This suggests that OurKD outperforms StKD in handling longer sentences, emphasizing the improved comprehension of semantic information in longer sentences after distillation with large-scale language teacher models.

Table 2: Comparison of different methods based on BLEU scores.

| Method | MN-CH | Kr-CH | JP-CH |
|--------|-------|-------|-------|
| AMTML-KD (Liu et al., 2020) | 38.1 | 36.7 | **37.9** |
| SKD (Liu et al., 2023a) | 38.5 | 36.3 | 37.1 |
| RMSKD (Yuan et al., 2021) | 38.2 | 37.3 | 37.1 |
| Multi-G (Liu et al., 2022) | 38.1 | 36.3 | 37.2 |
| RLAMKD (Our) | **38.6** | **36.8** | 37.8 |

Table 3 summarizes the BLEU score comparisons in ablation experiments conducted on Transformer, Performer, and Linformer models. These experiments encompass the comparison of performance with and without Multilingual Multi-Source Knowledge Distillation (MKD), Adversarial Perturbation (AP), Strander Knowledge Distillation (StKD) and Reinforcement Learning Agent (RLAgent) . The results demonstrate that MKD, RL Agent and AP have a significant impact on performance improvement. It is evident from this table that incorporating adversarial perturbation into knowledge distillation enhances model performance.

From table 2, we can observe that among various knowledge distillation methods, RLAMKD (our method) achieves the highest BLEU scores on both the MN-CH and Kr-CH datasets, demonstrating its superiority on these sets. While on the JP-CH dataset, AMTML-KD performs slightly better than RLAMKD, the score of RLAMKD is very close to the top, trailing by only 0.1 points. This indicates that RLAMKD exhibits consistent and impressive performance across all three datasets. Meanwhile, other methods like SKD, RMSKD, and Multi-G show relatively stable performance across the different datasets. Overall, RLAMKD appears to be the most effective among these methods, particularly on the MN-CH and Kr-CH datasets.

From Figure 4, after determining the optimal teacher model out of the four considered, we proceeded to combine different corpora to draw conclusions. It's evident that the BLEU scores vary depending on the corpora combination used. Notably, when considering the combination "1 & 3", specifically the "1 & 3 & 1 & 3" set, it exhibited the highest BLEU score of 38.62. This suggests that the translation quality is optimal when the En-Tu corpus is involved in this combination. On the contrary, the "1 & 2" combination, particularly the "1 & 2 & 1 & 2" set, yielded the lowest BLEU score of 38.55. Additionally, the "2 & 3 & 2 & 3" combination also achieved an impressive BLEU score of 38.61, which is almost on par with the highest score.

Finally, We conducted a detailed sensitivity analysis for each hyperparameter, and we also conducted ablation experiments for each component of the proposed reward function. The final results for all these experiments are presented in Appendices Table 5, 6, and 7.

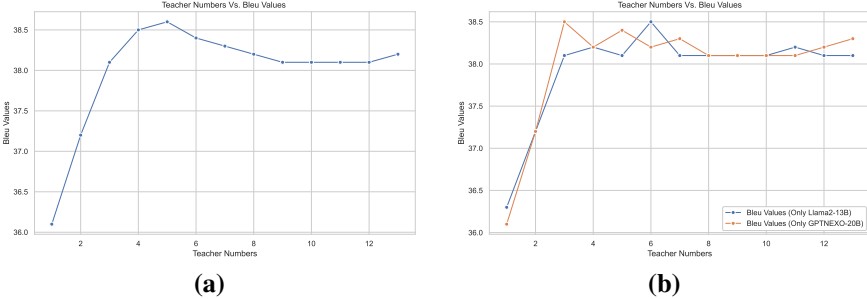

     **(a)**            **(b)**

Figure 2: (a) Depict the relationship between Teacher Numbers and Bleu Values (MN-CH). (b) Compare Bleu values for Llama2-13B and GPTNEXO-20B models as the number of teacher models (Teacher Numbers) increases( MN-CH).

## 4.4 DISCUSS

Based on the results mentioned above, we have observed that increasing the number of teacher models indeed enhances the model's performance. This further confirms the advantages of integrating

Table 3: Comparison of BLEU scores in ablation experiments on Transformer, Performer, and Linformer, including performance comparisons with and without Multilingual Multi-Source Knowledge Distillation (MKD), Adversarial Perturbation (AP), and Standard Knowledge Distillation (StKD). Reinforcement Learning Agent (OurRL)

| Method | Model | MN-CH | Kr-CH | JP-CH |
|---|---|---|---|---|
| | Performer(OurRL+AP) | **38.6** | **36.8** | **38.1** |
| | Transformer(OurRL+AP) | **38.3** | **36.8** | **38.2** |
| | Linformer(OurRL+AP) | **37.9** | **36.3** | **37.2** |
| | | | | |
| MKD | Performer(OurRL) | 37.8 | 35.3 | 37.1 |
| | Transformer(OurRL) | 37.6 | 34.9 | 37.2 |
| | Linformer(OurRL) | 37.2 | 35.1 | 37.1 |
| | | | | |
| | Performer(AP) | 36.8 | 35.1 | 35.7 |
| | Transformer(AP) | 36.2 | 34.9 | 35.6 |
| | Linformer(AP) | 36.1 | 34.9 | 35.4 |
| | | | | |
| | Performer(OurRL+AP) | 36.4 | 34.5 | 34.8 |
| | Transformer(OurRL+AP) | 36.4 | 34.2 | 34.2 |
| | Linformer(OurRL+AP) | 36.3 | 34.1 | 33.1 |
| | | | | |
| StKD | Performer(OurRL) | 36.2 | 34.2 | 34.3 |
| | Transformer(OurRL) | 36.2 | 34.2 | 34.4 |
| | Linformer(OurRL) | 36.3 | 34.1 | 33.1 |
| | | | | |
| | Performer(AP) | 36.2 | 34.1 | 33.7 |
| | Transformer(AP) | 36.2 | 33.9 | 32.3 |
| | Linformer(AP) | 36.5 | 33.7 | 33.3 |

**(c)**          **(d)**

Figure 3: (c) Compare Bleu scores for different training methods (EN-CH, EN-TR, RS-CH, Mix) with varying numbers of teacher models (2T, 5T, 8T) in MN-CH.(d) Compare OurKD and StKD (Standard KD) scores based on sentence length in MN-CH.

knowledge from multiple teacher models, especially in resource-constrained settings. Additionally, when compared to traditional reinforcement learning distillation, the distillation process employed in our study demonstrated superior performance, indirectly affirming the benefits of replacing the utility function. These findings are consistent with related research in the literature (Zhou et al., 2023; Dombry et al., 2023), and align well with the assumptions of our study.

Moreover, our research hypothesis is not solely focused on improving the performance of large language models; it also emphasizes the acquisition of knowledge. Through experiments, we have

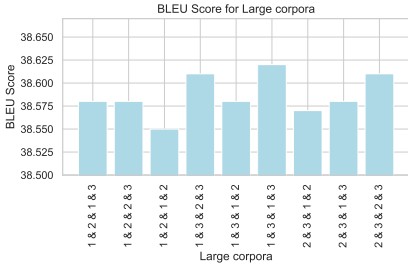

Figure 4: BLEU scores of different large corpora, where 1 represents En-Ch, 2 represents Ru-Ch, and 3 represents En-Tu.

successfully validated the validity of this hypothesis, which aligns with the findings of Zhao (Zhao et al., 2022).

Furthermore, the importance of teacher models trained on the EN-TU dataset for the Mongolian-Chinese translation task becomes particularly evident. This can be attributed to the presence of certain shared linguistic features between Mongolian and Chinese, similar to those between English and Turkish (Unseth, 2005). This suggests that knowledge derived from the EN-TU dataset offers valuable insights for Mongolian-Chinese translation.

Additionally, the emergence of large language models opens up a broader landscape for this type of knowledge transfer. Prior to the advent of large language models, capturing such nuanced cross-language transfers may have posed significant challenges. However, with the application of these models, we can not only identify such transfers but also delve deeper into investigating and harnessing them. This indirectly highlights the tremendous potential of large language models. As described in the literature (Liu et al., 2023b), these models' capabilities extend far beyond mere text generation or comprehension; they can function as versatile tools for a wide range of tasks, from basic text classification to intricate knowledge transfer and distillation. The extensive parameter space and comprehensive training data that these models utilize provide them with a wealth of linguistic and cultural knowledge, ensuring their exceptional performance in cross-language and cross-cultural tasks.

## 5 CONCLUSION AND FUTURE WORK

### 5.1 CONCLUSION

This work aims to address the challenges of low-resource Mongolian to Chinese machine translation by employing large-scale language models and multi-source knowledge distillation. Our primary contributions lie in significantly improving translation quality through effective knowledge transfer and the dynamic decision-making capabilities introduced by reinforcement learning (RL) agents. While empirical results have confirmed the effectiveness of this approach, we also acknowledge the substantial computational resources required. Additionally, there is potential for further optimization through the exploration of alternative RL algorithms in future research.

### 5.2 FUTURE WORK

Our future research will focus on enhancing the multi-source knowledge distillation process through contrastive learning and meta-learning techniques. We aim to extend our approach to other low-resource language pairs and NLP tasks, such as text generation and question-answering systems. We will also explore interdisciplinary approaches, including game theory, to further optimize our framework. Lastly, we are committed to ensuring that our research has a positive social impact, particularly for low-resource language communities.

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

## A  APPENDIX

### A.1  PROOF REWARD FUNCTION

#### A.1.1  REWARD METRIC IN REINFORCEMENT LEARNING FOR KNOWLEDGE DISTILLATION

In the context of knowledge distillation within reinforcement learning, the objective is to have the student model learn from the teacher model such that its output closely mirrors that of the teacher model. To quantify this closeness and provide feedback to the student model, we introduce the Continuous Ranked Probability Score (CRPS) as a reward metric (Hu et al., 2022).

Consider the outputs from both the student and teacher models. We interpret these outputs as Cu-mulative Distribution Functions (CDFs). CRPS serves as a metric that measures the discrepancy between two CDFs. It's specifically defined as:

$$\mathcal{R}_{\text{CRPS}}(\text{student}, \text{teacher}) = \frac{1}{T} \sum_{t=1}^{T} \int_0^1 \left[ F_t^{\text{student}}(p) - \varepsilon(p - F_t^{\text{teacher}}(p)) \right]^2 dp \qquad (16)$$

Here, $\varepsilon(x)$ is a step function, defined as:

$$\varepsilon(x) = \begin{cases} 0 & \text{if } x < 0 \\ 0.5 & \text{if } x = 0 \\ 1 & \text{if } x > 0 \end{cases} \tag{17}$$

The explanations for the other metric are as follows:

G(t) - Gini Coefficient: It is used to quantify the inequality or complexity among predictions made by the teacher model.

$$G(t) = 1 - \sum_{i=1}^{N} p_i^2 \tag{18}$$

where $p_i$ represents the probability distribution of different predictions by the teacher model at time $t$.

P(t) - Pearson Correlation Coefficient: It is employed to capture the consistency between the student and teacher models.

$$P(t) = \frac{\sum_{i=1}^{N}(x_i - \bar{x})(y_i - \bar{y})}{\sqrt{\sum_{i=1}^{N}(x_i - \bar{x})^2}\sqrt{\sum_{i=1}^{N}(y_i - \bar{y})^2}} \tag{19}$$

Here, $x_i$ represents predictions from the student model, $y_i$ represents predictions from the teacher model, and $\bar{x}$ and $\bar{y}$ are their respective means.

H(t) - Herfindahl Index**: It is used to assess the diversity among predictions from teacher models.

$$H(t) = \sum_{i=1}^{N} p_i^2 \tag{20}$$

where $p_i$ represents the probability distribution of different predictions by the teacher model at time $t$.

### A.1.2 Proof of Equivalence Between Discrete and Continuous Forms

**Lemma 1**:Transition from Discrete to Continuous

Consider a function $f(x)$ defined on the interval $[a, b]$. If we take a partition $P = \{x_0, x_1, \ldots, x_n\}$ within this interval (where $a = x_0 < x_1 < \ldots < x_n = b$), and select some point $c_i$ in each sub-interval $[x_{i-1}, x_i]$, then the Riemann sum is given by:

$$S(P, f) = \sum_{i=1}^{n} f(c_i)(x_i - x_{i-1}) \tag{21}$$

As the maximum length of the partition $P$ approaches zero, the Riemann sum converges to the integral of $f(x)$ over the interval $[a, b]$.

**Lemma 2**: Equivalence of Discrete and Continuous Forms

Assume we have a function $g(p)$ defined over $[0, 1]$. If we partition $[0, 1]$ into $n$ small equidistant intervals, each of length $\Delta p = \frac{1}{n}$, then:

$$\lim_{n \to \infty} \sum_{i=1}^{n} g(p_i)\Delta p = \int_0^1 g(p)dp \tag{22}$$

**Proof**:

Consider our continuous CRPS definition:

$$\mathcal{S}_{\text{CRPS}}(\text{student}, \text{teacher}) = \frac{1}{T} \sum_{t=1}^{T} \int_{0}^{1} \left[ F_t^{\text{student}}(p) - \varepsilon(p - F_t^{\text{teacher}}(p)) \right]^2 dp \tag{23}$$

Now, consider the discrete case. Suppose we partition the interval $[0, 1]$ into $n$ subintervals of equal length, each of length $\Delta p = \frac{1}{n}$. For each subinterval, we choose the midpoint $p_i$ as a representative and compute the difference at that point.

$$\mathcal{S}_{\text{CRPS,discrete}}(\text{student}, \text{teacher}) = \frac{1}{T} \sum_{t=1}^{T} \sum_{i=1}^{n} \left[ F_t^{\text{student}}(p_i) - \varepsilon(p_i - F_t^{\text{teacher}}(p_i)) \right]^2 \Delta p \tag{24}$$

Based on Lemma 2, as $n$ approaches infinity, the above discrete form will converge to the continuous form. This establishes the equivalence between the discrete and continuous forms of CRPS.

### A.1.3 CONTINUITY AND DIFFERENTIABILITY

**Lemma 3**: Square of the Difference of Two Continuous Functions

Assume $f(p)$ and $g(p)$ are continuous functions defined over the interval $[0, 1]$. Then the function $h(p) = [f(p) - g(p)]^2$ is also continuous.

Proof: Since $f$ and $g$ are both continuous, their difference $f(p) - g(p)$ is also continuous. The square of a continuous function remains continuous over its domain. Therefore, $h(p)$ is continuous.

**Lemma 4**: Differentiability of Continuous Functions

Suppose $f(p)$ is a continuous function defined over $[0, 1]$ and is differentiable almost everywhere on this interval. Then its integral $\int_0^1 f(p)dp$ exists and is continuous.

Proof: The integral of a continuous function always exists over its domain. Since $f(p)$ is differentiable almost everywhere on $[0, 1]$, its antiderivative (indefinite integral) is continuous. Therefore, its definite integral (over a specific interval) is also continuous.

**Proof**:

Consider our CRPS definition:

$$\mathcal{S}_{\text{CRPS}}(\text{student}, \text{teacher}) = \frac{1}{T} \sum_{t=1}^{T} \int_{0}^{1} \left[ F_t^{\text{student}}(p) - \varepsilon(p - F_t^{\text{teacher}}(p)) \right]^2 dp \tag{25}$$

According to Lemma 3, the squared function $[F_t^{\text{student}}(p) - \varepsilon(p - F_t^{\text{teacher}}(p))]$ is continuous over $[0, 1]$.

As this function is continuous over $[0, 1]$ and since the antiderivative of a continuous function is continuous, we can deduce that the CRPS formula is continuous over $[0, 1]$.

Regarding differentiability, since CRPS is a function of $[F_t^{\text{student}}(p)]$ and $[F_t^{\text{teacher}}(p)]$, we only need to prove that these two functions are differentiable almost everywhere over $[0, 1]$. Since they are both CDFs, they are continuous over $[0, 1]$ and are differentiable almost everywhere. Therefore, based on Lemma 4, the CRPS formula is continuous and differentiable almost everywhere over $[0, 1]$.

### A.2 TRAIN CONFIGURATION

In this experiment, we chose Python (3.8.1) as our programming language and utilized the Pytorch (2.6.11) deep learning framework. The model weights were initialized using a random initialization method. To prevent overfitting, we employed a dropout rate of 0.5 and implemented Layer Normalization. Furthermore, we primarily used the ReLU activation function. Whenever the validation performance reached a new peak, we saved the model weights. All computations were conducted on a system equipped with 4 NVIDIA A100 GPUs and 64GB RAM.

Table 4: The impact of setting different loss functions during distillation on the final results

| Loss | BLEU |
|------|------|
| MSE | 37.4 |
| CrossEntropy | 37.8 |
| MSE+CrossEntropy | **38.6** |

### A.3 REASON FOR TEACHER TRAIN DATASET

- English-Chinese: There are a total of 17.8 million bilingual sentence pairs, which are part of the language data provided by WMT17.

- English-Turkish: There are a total of 28.3 million bilingual sentence pairs, and the data source is based on the work referenced in (Yirmibeşoğlu et al., 2023).

- Russian-Chinese: There are a total of 63.2 million bilingual sentence pairs, and the data source is based on the work referenced in (Hasebe, 2015).

The main reason for selecting English-Chinese and Russian-Chinese corpora as the training data for the teacher model is that these language pairs constitute extensive corpora covering major languages from both Europe and Asia. These corpora provide high-quality data that can significantly facilitate the model's training (Jooste et al., 2022). Additionally, the diversity of these languages, belonging to different language families, and having distinct structures, grammar rules, and expressions, aids the teacher model in better learning and comprehension, which in turn enhances its performance in multilingual scenarios.

The choice of English-Turkish as the training data for the multi-source knowledge distillation teacher model, used to distill the Mongolian language student model, is primarily due to the fact that Mongolian and Turkish belong to the same Altaic language family and share many similarities (Shi et al., 2008). It is hoped that languages from similar language families can distill some shared features.

### A.4 DATASETS DETAIL

- Mongolia-Chinese (MN-CH): 8.7 million sentences in Mongolia and Chinese,We allocated 30% of the data for validation and the remaining 70% for training.

- Korean-Chinese (Kr- CH) (Park et al., 2020): 12.5 million sentences in Korean and Chinese, We allocated 30% of the data for validation and the remaining 70% for training.

- Japanese-Chinese (JP-CH) (Zhang et al., 2022): 12 million sentences in Japaneseand Chinese,We allocated 30% of the data for validation and the remaining 70% for training.

### A.5 TRAINING

The figures and tables presented in this section provide a comprehensive overview of the experiment results. In Figures 5 and 6, we track various metrics during the training and validation phases, including BLEU scores and loss functions. Figure 5(e) displays the BLEU score trends during training and validation, highlighting the model's performance over time. Figure 5(f) compares BLEU scores among different models during validation, offering insights into their relative effectiveness. Figures 6 (g) and (h) illustrate the behavior of the loss functions during training and validation, providing a visual representation of the optimization process. Figure 7 illustrates the reward trends throughout the entire distillation process. The x-axis in the graph represents time or different stages of distillation, while the y-axis represents the reward values. In this graph, we can observe that the reward values consistently increase over the course of the entire distillation phase.

Additionally, Table 4 examines the impact of different loss functions on the final BLEU scores, demonstrating that combining MSE and CrossEntropy yields the best result. Furthermore, Table 5 explores the sensitivity of the hyperparameter $\beta$, showcasing how variations in its value affect the BLEU scores.

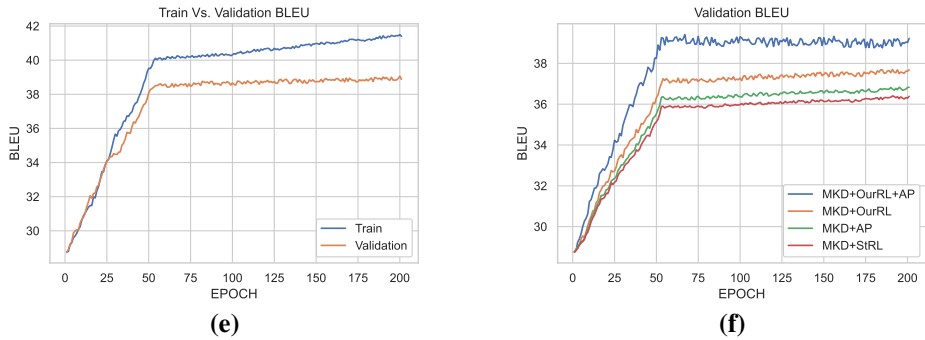

Figure 5: (e) BLEU score during training and validation. (f) Comparing BLEU scores among different models during validation.

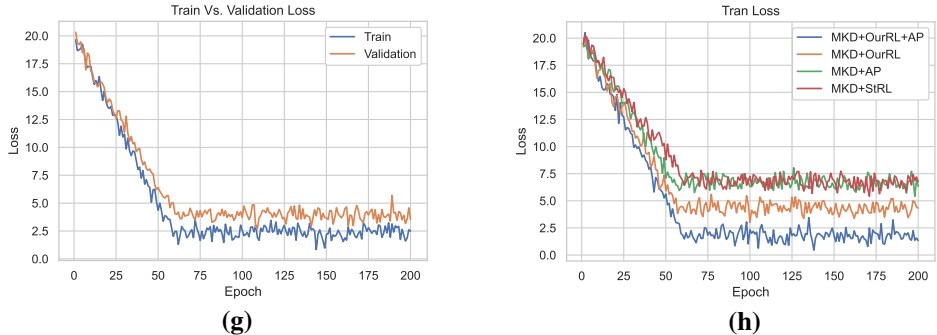

Figure 6: (g) Loss function during training and validation. (h) Comparing loss functions among different models during validation.

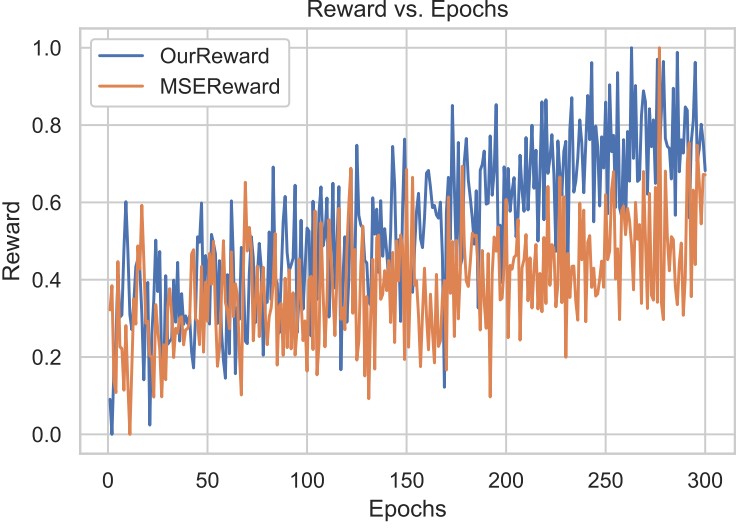

Figure 7: Reward Trends

Table 5: Sensitivity Test of $\beta$

| Value | BLEU |
|-------|------|
| 0.50 | 36.2 |
| 0.51 | 36.3 |
| 0.53 | 36.5 |
| 0.55 | 36.7 |
| 0.57 | 36.9 |
| 0.58 | 37.0 |
| 0.59 | 37.1 |
| 0.61 | 37.2 |
| 0.63 | 37.4 |
| 0.65 | 37.5 |
| 0.66 | 37.3 |
| 0.67 | 37.2 |
| 0.68 | 37.1 |
| 0.69 | 37.0 |
| 0.70 | 38.6 |
| 0.71 | 38.5 |
| 0.72 | 38.4 |
| 0.73 | 38.3 |
| 0.74 | 38.2 |
| 0.75 | 38.3 |
| 0.76 | 38.2 |
| 0.77 | 38.2 |

Table 6: Sensitivity Test of $\alpha$

| Value | BLEU |
|-------|------|
| 0.10 | 37.2 |
| 0.20 | 38.3 |
| 0.30 | 38.5 |
| 0.32 | 38.6 |
| 0.34 | 38.2 |
| 0.36 | 38.1 |
| 0.38 | 38.0 |
| 0.40 | 37.8 |
| 0.50 | 37.8 |

## A.6 MULTIPLE-SOURCE KNOWLEDGE DISTILLATION WITH REINFORCEMENT LEARNING AND ADVERSARIAL TRAINING ALGORITHM DESCRIPTION

These parameters and variables play different roles in the pseudo-code:

- $s$ represents the current state, used to describe the context in reinforcement learning.

- $r_t$ is the reward function used to evaluate the performance of the student model. It calculates rewards based on the comparison between the student model's outputs and reference translations to guide the reinforcement learning process.

- $\beta$ is a hyperparameter that represents the weight of the adversarial loss in the total loss, controlling the strength of adversarial training.

- $N_{\text{epochs}}$ indicates the total number of training epochs, i.e., how many iterations the entire training process will go through.

Table 7: Reward Components and BLEU Values in $\alpha = 0.32$

| Reward Components | BLEU |
|---|---|
| + $\mathcal{S}_{\text{CRPS}}$ | 38.1 |
| + GINI | 36.1 |
| + P | 36.3 |
| + H | 36.2 |
| + BLEU | 36.8 |
| + $\mathcal{S}_{\text{CRPS}}$ + GINI | 38.2 |
| + $\mathcal{S}_{\text{CRPS}}$ + P | 38.2 |
| + $\mathcal{S}_{\text{CRPS}}$ + P + H | 38.3 |
| + $\mathcal{S}_{\text{CRPS}}$ + P + H + BLEU | 38.4 |
| + $\mathcal{S}_{\text{CRPS}}$ + BLEU | 38.4 |
| + $\mathcal{S}_{\text{CRPS}}$ + P + BLEU | 38.2 |
| + $\mathcal{S}_{\text{CRPS}}$ + H + BLEU | 38.3 |
| + $\mathcal{S}_{\text{CRPS}}$ + GINI + BLEU | 38.1 |
| + MSE + BLEU | 37.3 |
| + MSE + GINI + P + H + BLEU | 37.4 |
| + MSE | 37.2 |
| + $\mathcal{S}_{\text{CRPS}}$ + GINI + P + H + BLEU | **38.6** |

- $D$ represents the training dataset, which includes data batches used to train the student model.

These parameters and variables serve different purposes in the context of multiple-source knowledge distillation and reinforcement learning, helping control the training process, choose actions, calculate rewards, and evaluate performance.

---

**Algorithm 1** Multiple-Source Knowledge Distillation with Reinforcement Learning and Adversarial Training

---

1: **procedure** RLAMKD
2:     Initialize the agent: agent $\leftarrow$ Agent(num_teachers)
3:     **for** epoch $\leftarrow$ 1 to $N_{\text{epochs}}$ **do**
4:         **for** each d sample data in the $D$ **do**
5:             Assign weights to teacher models: weights_ts $\leftarrow$ agent.weights_teachers()
6:             Calculate $loss$: loss $\leftarrow$ compute_kd_ad_loss(d, weights_ts, t_models, s_model, $\beta$)
7:             Calculate $r_t$: $r_t \leftarrow$ compute_reward(d, s_model, t_models)
8:             Get state: agent.get_state(d, s_model, t_models)
9:             Update agent's weights: agent.update_weights(weights_ts, reward_value)
10:         **end for**
11:     **end for**
12: **end procedure**

---

