# OpenReview forum: "Maximizing LLMs Potential: Enhancing Mongolian Chinese Machine Translation with RL Agents and Adversarial Multi Knowledge Distillation"
_ICLR.cc/2024/Conference — ICLR 2024 Conference Withdrawn Submission_

### Official Review · Reviewer_Mjnh · 2023-10-21

**Soundness:** 1 poor
**Presentation:** 1 poor
**Contribution:** 2 fair
**Rating:** 1
**Confidence:** 5

**Summary:**

This paper proposes an innovative approach that combines multi-source knowledge distillation and incorporates Reinforcement
Learning (RL) to help models acquire and transfer knowledge from LLMs more effectively. RL plays a crucial role in this, making dynamic decisions to determine useful information for low-resource translation models and how to extract it efficiently. They introduce a new reward function to comprehensively guide knowledge distillation and experiments show that this approach harnesses the potential
of LLMs, significantly improving translation quality in low-resource settings.

**Strengths:**

- This paper notices that the LLMs' potential in low-resource NMT
- This paper tries multi-source distillation with RL method, it has somewhat novelty

**Weaknesses:**

- This paper is hard to read. In other words, its presentation is very poor. It seems that the authors do not spend enough time to prepare such a paper. Specifically, "RL" in the title; what is the test sets and setting on Table 1, why the results of ChatGPT or GPT4 are not presented? What is the i of M_t^i? What is the M_y. What do you mean by "follow 11", above Equation 11?  What is "eq.3"
- This paper cannot be viewed as a solid science paper, as I cannot find enough information to ensure their experiments are reasonable and convincing. What is your test set in Table 2, are they officially released test sets?  Why the student model's lr is lower than the teacher model? How do you train the LLMs, i.e., the teacher models? Do you mean that you will SFT them with millions or billions datas?
- The experiments in this paper are not enough, and miss some important baselines.

**Questions:**

1 See above
2 Why not put Figure 1 stick with its description in the main content?
3 Why so many duplicate contents in model selection?
4 Why not cite ChatGPT or GPT4 related paper rather than GPT3 in the first line?

---

### Official Review · Reviewer_NGkR · 2023-10-30

**Soundness:** 2 fair
**Presentation:** 3 good
**Contribution:** 2 fair
**Rating:** 5
**Confidence:** 3

**Summary:**

This paper has introduced a novel framework for multi-source distillation in the context of machine translation, employing reinforcement learning. To be specific, this framework seamlessly integrates three distinct components within a cohesive distillation strategy. The student model initially acquires information from various language models, each trained with diverse datasets, and subsequently undergoes training with the adaptive decision-making capabilities of an RL agent to enhance its overall performance. Additionally, a novel reward function is introduced, meticulously accounting for the intricate and multifaceted decision-making processes. Finally, this paper presents empirical results in the context of low-resource Mongolian-to-Chinese translation.

**Strengths:**

- The paper is easily comprehensible and straightforward.
- The authors conduct an ablation study to systematically assess the contributions of each individual component.

**Weaknesses:**

- The observed improvements are relatively marginal when compared to other baseline methods.
- The paper lacks a comprehensive set of experiments to conclusively establish the superiority of the proposed approach over other existing methods. The performance enhancement is primarily demonstrated through improvements in BLEU scores in various translation directions. However, I'm hard to find a qualitative assessment of the method's effectiveness and why the proposed techniques are advantageous in enhancing translation models. Since the authors mentioned that the consideration of dynamic changes in the distillation process is a key contribution of the paper, it would be good to provide additional experiments to demonstrate this.

**Questions:**

- The experimental results appear to exhibit a marginal improvement.
    -  Are these results(Table 2) indicative of the average outcomes achieved through multiple runs, and has statistical significance been established regarding the variance in performance between the baseline and the proposed model?

- Does the performance trend change depending on the student model size?

---

### Official Review · Reviewer_hubM · 2023-11-02

**Soundness:** 3 good
**Presentation:** 3 good
**Contribution:** 2 fair
**Rating:** 5
**Confidence:** 5

**Summary:**

In this paper, the authors enhance the performance of the  low-resource Mongolian-to-Chinese machine translation by integrating multi-source knowledge distillation and reinforcement learning in large language model. Specfically, the authors propose a new reward function designed to guide the knowledge distillation process and introduce adversarial noise to improve the effectiveness of the knowledge distillation process.

**Strengths:**

1. The authors proposed to use LLMs as the teachter model to distill the task-specific models, this is a promising direction for achieving a balance between performance and efficiency.

2. Experimental results in multiple translation tasks show the effectiveness of the proposed method.

**Weaknesses:**

1. Why have you selected Llama as one of the teacher models? Llama has not been trained on the languages referenced in the paper.
2. I want to know the performance of the reward models. In recent work, the performance of the reward model determines the final outcome.
3. Training a model based on Reinforcement Learning (RL) is challenging. Is it possible to directly use rejection sampling for knowledge distillation?

**Questions:**

See Weaknesses

---

### Meta-Review · Area_Chair_qDnc · 2023-12-02

**Metareview:**

The paper under review presents an approach to enhance machine translation performance, particularly focusing on the Mongolian-to-Chinese language pair, by integrating multi-source knowledge distillation and reinforcement learning into LLMs.

The paper is criticized for poor presentation and lack of clarity, with several ambiguities and missing details in the methodology and results. The paper lacks comprehensive experiments to establish its superiority conclusively over existing methods. The current paper includes some unreasonable settings. However, the authors do not provide any reply to the above comments.

**Justification For Why Not Higher Score:**

As mentioned in the meta-review, the paper has several areas that need to be improved.

**Justification For Why Not Lower Score:**

N/A

---

### Decision · Program_Chairs · 2024-01-16

Reject